# Adversarial Attacks on Adversarial Bandits

**Yuzhe Ma**
Microsoft Azure AI
yuzhema@microsoft.com

**Zhijin Zhou**[*]
Amazon
zhijin@amazon.com

## Abstract

We study a security threat to adversarial multi-armed bandits, in which an attacker perturbs the loss or reward signal to control the behavior of the victim bandit player. We show that the attacker is able to mislead any no-regret adversarial bandit algorithm into selecting a suboptimal target arm in every but sublinear $(T - o(T))$ number of rounds, while incurring only sublinear $(o(T))$ cumulative attack cost. This result implies critical security concern in real-world bandit-based systems, e.g., in online recommendation, an attacker might be able to hijack the recommender system and promote a desired product. Our proposed attack algorithms require knowledge of only the regret rate, thus are agnostic to the concrete bandit algorithm employed by the victim player. We also derived a theoretical lower bound on the cumulative attack cost that any victim-agnostic attack algorithm must incur. The lower bound matches the upper bound achieved by our attack, which shows that our attack is asymptotically optimal.

## 1 Introduction

Multi-armed bandit presents a sequential learning framework that enjoys applications in a wide range of real-world domains, including medical treatment Zhou et al. (2019); Kuleshov & Precup (2014), online advertisement Li et al. (2010), resource allocation Feki & Capdevielle (2011); Whittle (1980), search engines Radlinski et al. (2008), etc. In bandit-based applications, the learning agent (bandit player) often receives reward or loss signals generated through real-time interactions with users. For example, in search engine, the user reward can be clicks, dwelling time, or direct feedbacks on the displayed website. The user-generated loss or reward signals will then be collected by the learner to update the bandit policy. One security caveat in user-generated rewards is that there can be malicious users who generate adversarial reward signals. For instance, in online recommendation, adversarial customers can write fake product reviews to mislead the system into making wrong recommendations. In search engine, cyber-attackers can create click fraud through malware and causes the search engine to display undesired websites. In such cases, the malicious users influence the behavior of the underlying bandit algorithm by generating adversarial reward data. Motivated by that, there has been a surge of interest in understanding potential security issues in multi-armed bandits, i.e., to what extend are multi-armed bandit algorithms susceptible to adversarial user data.

Prior works mostly focused on studying reward attacks in the stochastic multi-armed bandit setting Jun et al. (2018); Liu & Shroff (2019), where the rewards are sampled according to some distribution. In contrast, less is known about the vulnerability of adversarial bandits, a more general bandit framework that relaxes the statistical assumption on the rewards and allows arbitrary (but bounded) reward signals. The adversarial bandit also has seen applications in a broad class of real-world problems especially when the reward structure is too complex to model with a distribution, such as inventory control Even-Dar et al. (2009) and shortest path routing Neu et al. (2012). Similarly, the same security problem could arise in adversarial bandits due to malicious users. Therefore, it is imperative to investigate potential security caveats in adversarial bandits, which provides insights to help design more robust adversarial bandit algorithms and applications.

In this paper, we take a step towards studying reward attacks on adversarial multi-armed bandit algorithms. We assume the attacker has the ability to perturb the reward signal, with the goal of misleading the bandit algorithm to always select a target (sub-optimal) arm desired by the attacker. Our

---

[*]This work does not relate to the author's position at Amazon, no matter how the author affiliates.

main contributions are summarized as below. (1) We present attack algorithms that can successfully force arbitrary no-regret adversarial bandit algorithms into selecting any target arm in $T - o(T)$ rounds while incurring only $o(T)$ cumulative attack cost, where $T$ is the total rounds of bandit play. (2) We show that our attack algorithm is theoretically optimal among all possible *victim-agnostic* attack algorithms, which means that no other attack algorithms can successfully force $T - o(T)$ target arm selections with a smaller cumulative attack cost than our attacks while being agnostic to the underlying victim bandit algorithm. (3) We empirically show that our proposed attack algorithms are efficient on both vanilla and a robust version of Exp3 algorithm Yang et al. (2020).

## 2 PRELIMINARIES

The bandit player has a finite action space $\mathcal{A} = \{1, 2, ..., K\}$, where $K$ is the total number of arms. There is a fixed time horizon $T$. In each time step $t \in [T]$, the player chooses an arm $a_t \in \mathcal{A}$, and then receives loss $\ell_t = \mathcal{L}_t(a_t)$ from the environment, where $\mathcal{L}_t$ is the loss function at time $t$. In this paper, we consider "loss" instead of reward, which is more standard in adversarial bandits. However, all of our results would also apply in the reward setting. Without loss of generality, we assume the loss functions are bounded: $\mathcal{L}_t(a) \in [0, 1], \forall a, t$. Moreover, we consider the so-called non-adaptive environment Slivkins (2019); Bubeck & Cesa-Bianchi (2012), which means the loss functions $\mathcal{L}_{1:T}$ are fixed beforehand and cannot change adaptively based on the player behavior after the bandit play starts. The goal of the bandit player is to minimize the difference between the cumulative loss incurred by always selecting the optimal arm in hindsight and the cumulative loss incurred by the bandit algorithm, which is defined as the regret below.

**Definition 2.1.** *(Regret). The regret of the bandit player is*

$$R_T = \mathbf{E}\left[\sum_{t=1}^{T} \mathcal{L}_t(a_t)\right] - \min_a \sum_{t=1}^{T} \mathcal{L}_t(a), \tag{1}$$

*where the expectation is with respect to the randomness in the selected arms $a_{1:T}$.*

We now make the following major assumption on the bandit algorithm throughout the paper.

**Assumption 2.2.** *(No-regret Bandit Algorithm). We assume the adversarial bandit algorithm satisfies the "no-regret" property asymptotically, i.e., $R_T = O(T^\alpha)$ for some $\alpha \in [\frac{1}{2}, 1)$[1].*

As an example, the classic adversarial bandit algorithm Exp3 achieves $\alpha = \frac{1}{2}$. In later sections, we will propose attack algorithms that apply not only to Exp3, but also arbitrary no-regret bandit algorithms with regret rate $\alpha \in [\frac{1}{2}, 1)$. Note that the original loss functions $\mathcal{L}_{1:T}$ in (1) could as well be designed by an adversary, which we refer to as the "environmental adversary". In typical regret analysis of adversarial bandits, it is implicitly assumed that the environmental adversary aims at inducing large regret on the player. To counter the environmental adversary, algorithms like Exp3 introduce randomness into the arm selection policy, which provably guarantees sublinear regret for arbitrary sequence of adversarial loss functions $\mathcal{L}_{1:T}$.

### 2.1 MOTIVATION OF ATTACKS ON ADVERSARIAL BANDITS

In many bandit-based applications, an adversary may have an incentive to pursue different attack goals than boosting the regret of the bandit player. For example, in online recommendation, imagine the situation that there are two products, and both products can produce the maximum click-through rate. We anticipate a fair recommender system to treat these two products equally and display them with equal probability. However, the seller of the first product might want to mislead the recommender system to break the tie and recommend his product as often as possible, which will benefit him most. Note that even if the recommender system chooses to display the first product every time, the click-through rate (i.e., reward) of the system will not be compromised because the first product has the maximum click-through rate by assumption, thus there is no regret in always recommending it. In this case, misleading the bandit player to always select a target arm does not boost the regret. We point out that in stochastic bandits, forcing the bandit player to always select a sub-optimal target arm must

---

[1]We assume $\alpha \geq \frac{1}{2}$ because prior works Auer et al. (1995); Gerchinovitz & Lattimore (2016) have proved that the regret has lower bound $\Omega(\sqrt{T})$.

induce linear regret. Therefore, a robust stochastic bandit algorithm that recovers sublinear regret in presence of an attacker can prevent a sub-optimal arm from being played frequently. However, in adversarial bandit, the situation is fundamentally different. As illustrated in example 1, always selecting a sub-optimal target arm may still incur sublinear regret. As a result, robust adversarial bandit algorithms that recover sublinear regret in presence of an adversary (e.g., Yang et al. (2020)) can still suffer from an attacker who aims at promoting a target arm.

**Example 1.** *Assume there are $K = 2$ arms $a_1$ and $a_2$, and the loss functions are as below.*

$$\forall t, \mathcal{L}_t(a) = \left\{ \begin{array}{ll} 1 - \sqrt{T}/T & \text{if } a = a_1, \\ 1 & \text{if } a = a_2. \end{array} \right. \tag{2}$$

*Note that $a_1$ is the best-in-hindsight arm, but always selecting $a_2$ induces $\sqrt{T}$ regret, which is sublinear and does not contradict the regret guarantee of common bandit algorithms like Exp3.*

## 3 THE ATTACK PROBLEM FORMULATION

While the original loss functions $\mathcal{L}_{1:T}$ can already be adversarial, an adversary who desires a target arm often does not have direct control over the environmental loss functions $\mathcal{L}_{1:T}$ due to limited power. However, the adversary might be able to perturb the instantiated loss value $\ell_t$ slightly. For instance, a seller cannot directly control the preference of customers over different products, but he can promote his own product by giving out coupons. To model this attack scenario, we introduce another adversary called the "attacker", an entity who sits in between the environment and the bandit player and intervenes with the learning procedure. We now formally define the attacker in detail.

(Attacker Knowledge). We consider an (almost) black-box attacker who has very little knowledge of the task and the victim bandit player. In particular, the attacker does ***not*** know the clean environmental loss functions $\mathcal{L}_{1:T}$ beforehand. Furthermore, the attacker does ***not*** know the concrete bandit algorithm used by the player. However, the attacker knows the regret rate $\alpha$[2].

(Attacker Ability) In each time step $t$, the bandit player selects an arm $a_t$ and the environment generates loss $\ell_t = \mathcal{L}_t(a_t)$. The attacker sees $a_t$ and $\ell_t$. Before the player observes the loss, the attacker has the ability to perturb the original loss $\ell_t$ to $\tilde{\ell}_t$. The player then observes the perturbed loss $\tilde{\ell}_t$ instead of the original loss $\ell_t$. The attacker, however, cannot arbitrarily change the loss value. In particular, the perturbed loss $\tilde{\ell}_t$ must also be bounded: $\tilde{\ell}_t \in [0, 1], \forall t$.

(Attacker Goal). The goal of the attacker is two-fold. First, the attacker has a desired target arm $a^\dagger$, which can be some sub-optimal arm. The attacker hopes to mislead the player into selecting $a^\dagger$ as often as possible, i.e., maximize $N_T(a^\dagger) = \sum_{t=1}^T \mathbb{1}\left[a_t = a^\dagger\right]$. On the other hand, every time the attacker perturbs the loss $\ell_t$, an attack cost $c_t = |\tilde{\ell}_t - \ell_t|$ is induced. The attacker thus hopes to achieve a small cumulative attack cost over time, defined as below.

**Definition 3.1.** *(Cumulative Attack Cost). The cumulative attack cost of the attacker is defined as*

$$C_T = \sum_{t=1}^T c_t, \text{ where } c_t = |\tilde{\ell}_t - \ell_t|. \tag{3}$$

The focus of our paper is to design efficient attack algorithms that can achieve $\mathbf{E}\left[N_T(a^\dagger)\right] = T - o(T)$ and $\mathbf{E}\left[C_T\right] = o(T)$ while being ***agnostic*** to the concrete victim bandit algorithms.

Intuitively, if the total loss of the target arm $\sum_{t=1}^T \mathcal{L}_t(a^\dagger)$ is small, then the attack goals would be easy to achieve. In the extreme case, if $\mathcal{L}_t(a^\dagger) = 0, \forall t$, then even without attack, $a^\dagger$ is already the optimal arm and will be selected frequently in most scenarios [3]. On the other hand, if $\mathcal{L}_t(a^\dagger) = 1, \forall t$, then the target arm is always the worst arm, and forcing the bandit player to frequently select $a^\dagger$ will require the attacker to significantly reduce $\mathcal{L}_t(a^\dagger)$. In later sections, we will formalize this intuition and characterize the attack difficulty.

---

[2]It suffices for the attacker to know an upper bound on the regret rate to derive all the results in our paper, but for simplicity we assume the attacker knows exactly the regret rate.

[3]An exceptional case is when there exists some non-target arm $a'$ that also has 0 loss in every round, then $a'$ is equally optimal as $a^\dagger$, and without attack $a'$ will be selected equally often as $a^\dagger$.

# 4 ATTACK WITH TEMPLATE LOSS FUNCTIONS

In this section, we first propose a general attack strategy called "template-based attacks". The template-based attacks perform loss perturbations according to a sequence of template loss functions $\tilde{\mathcal{L}}_{1:T}$. The templates $\tilde{\mathcal{L}}_{1:T}$ are determined before the bandit play starts. Then in each time step $t$ during the bandit play, the attacker perturbs the original loss $\ell_t$ to $\tilde{\ell}_t = \tilde{\mathcal{L}}_t(a_t)$. Note that template-based attacks may seem weak at first glance, because the template loss functions are fixed beforehand and thus non-adaptive to the behaviors of the victim bandit player. This is in stark contrast to most prior works such as Jun et al. (2018). However, as we will show in later sections, template-based attacks are efficient and can even achieve the optimal attack cost.

We first make the following important observation, which is a critical property used to prove the main theoretical results in our paper.

**Observation 1.** *(Equivalence of Attack) Due to the partial observability of loss functions in the multi-armed bandit framework, running any bandit algorithm in the original environment $\mathcal{L}_{1:T}$ with template-based attack $\tilde{\mathcal{L}}_{1:T}$, is equivalent to, running the same algorithm in an environment with loss functions $\tilde{\mathcal{L}}_{1:T}$. In particular, the standard regret guarantee $R_T = O(T^\alpha)$ holds with respect to the template loss functions $\tilde{\mathcal{L}}_{1:T}$.*

We next instantiate the template-based attack on an easy attack scenario and general attack scenarios. For each scenario, we will design the template loss functions $\tilde{\mathcal{L}}_{1:T}$ ,

## 4.1 AN EASY ATTACK SCENARIO

In this section, we first study a relatively easy attack scenario, which serves as a motivating example that illustrates two key principles behind the design of our attack algorithms: (1) *Try to keep the loss of the target arm $\mathcal{L}_t(a^\dagger)$ unchanged*; and (2) *Boost the loss of all the other non-target arms to the maximum*. The easy attack scenario has the following assumption on the original loss functions, which requires the loss of the target arm to be at least $\rho$ bounded away from the maximum value.

**Assumption 4.1.** *(Easy Attack Scenario). There exists some constant $\rho \in (0, 1]$ such that*

$$\mathcal{L}_t(a^\dagger) \in [0, 1 - \rho], \forall t \in [T]. \tag{4}$$

The boundedness condition (4) needs to hold over all $T$ rounds. If assumption 4.1 holds, then the attacker can design the template loss functions $\tilde{\mathcal{L}}_t$ as in (5) to perform attack.

$$\forall t, \tilde{\mathcal{L}}_t(a) = \begin{cases} \mathcal{L}_t(a) & \text{if } a = a^\dagger, \\ 1 & \text{otherwise.} \end{cases} \tag{5}$$

**Remark 4.2.** *A few remarks are in order. First, note that although the form of $\tilde{\mathcal{L}}_t(a)$ depends on $\mathcal{L}_t(a)$, the attacker does not require knowledge of the original loss functions $\mathcal{L}_{1:T}$ beforehand to implement the attack. This is because when $a_t = a^\dagger$, the perturbed loss is $\tilde{\ell}_t = \tilde{\mathcal{L}}_t(a_t) = \mathcal{L}_t(a^\dagger) = \ell_t$ while $\ell_t$ is observable. When $a_t \neq a^\dagger$, $\tilde{\ell}_t$ can be directly set to 1. Second, note that the target arm $a^\dagger$ becomes the best-in-hindsight arm after attack. Consider running a no-regret bandit algorithm on the perturbed loss $\tilde{\mathcal{L}}_{1:T}$, since $\tilde{\mathcal{L}}_t(a^\dagger) = \mathcal{L}_t(a^\dagger) \leq 1 - \rho$, every time the player selects a non-target arm $a_t \neq a^\dagger$, it will incur at least $\rho$ regret. However, the player is guaranteed to achieve sublinear regret on $\tilde{\mathcal{L}}_{1:T}$ by observation 1, thus non-target arms can at most be selected in sublinear rounds. Finally, note that the loss remains unchanged when the target arm $a^\dagger$ is selected. This design is critical because should the attack be successful, then $a^\dagger$ will be selected in $T - o(T)$ rounds. By keeping the loss of the target arm unchanged, the attacker does not incur attack cost when the target arm is selected. As a result, our design (5) induces sublinear cumulative attack cost.*

**Theorem 4.3.** *Assume assumption 4.1 holds, and the attacker applies (5) to perform attack. Then there exists a constant $M > 0$ such that the expected number of target arm selections satisfies $\mathbf{E}\left[N_T(a^\dagger)\right] \geq T - MT^\alpha/\rho$, and the expected cumulative attack cost satisfies $\mathbf{E}\left[C_T\right] \leq MT^\alpha/\rho$.*

**Remark 4.4.** *Note that as the regret rate $\alpha$ decreases, the target arm selections $\mathbf{E}\left[N_T(a^\dagger)\right]$ increases and the cumulative attack cost $\mathbf{E}\left[C_T\right]$ reduces. That means, our attack algorithm becomes more*

*effective and efficient if the victim bandit algorithm has a better regret rate. The constant $M$ comes from the regret bound of the victim adversarial bandit algorithm and will depend on the number of arms $K$ (similarly for Theorem 4.6 and 4.9). We do not spell out its concrete form here because our paper aims at designing general attacks against arbitrary adversarial bandit algorithms that satisfy assumption 2.2. The constant term in the regret bound may take different forms for different algorithms. Comparatively, the sublinear regret rate $\alpha$ is more important for attack considerations.*

## 4.2 GENERAL ATTACK SCENARIOS

Our analysis in the easy attack scenario relies on the fact that every time the player fails to select the target arm $a^\dagger$, at least a constant regret $\rho$ will be incurred. Therefore, the player can only take non-target arms sublinear times. However, this condition breaks if there exists time steps $t$ where $\mathcal{L}_t(a^\dagger) = 1$. In this section, we propose a more generic attack strategy, which provably achieves sublinear cumulative attack cost on any loss functions $\mathcal{L}_{1:T}$. Furthermore, the proposed attack strategy can recover the result of Theorem 4.3 (up to a constant) when it is applied in the easy attack scenario. Specifically, the attacker designs the template loss functions $\tilde{\mathcal{L}}_t$ as in (6) to perform attack.

$$\forall t, \tilde{\mathcal{L}}_t(a) = \begin{cases} \min\{1 - t^{\alpha+\epsilon-1}, \mathcal{L}_t(a)\} & \text{if } a = a^\dagger, \\ 1 & \text{otherwise,} \end{cases} \quad (6)$$

where $\epsilon \in [0, 1 - \alpha)$ is a free parameter chosen by the attacker. We discuss how the parameter $\epsilon$ affects the attack performance in remark 4.7.

**Remark 4.5.** *Similar to (5), the attacker does not require knowledge of the original loss functions $\mathcal{L}_{1:T}$ beforehand to implement the attack. When a non-target arm is selected, the attacker always increases the loss to the maximum value 1. On the other hand, when the target arm $a^\dagger$ is selected, then if the observed clean loss value $\ell_t = \mathcal{L}_t(a_t) > 1 - t^{\alpha+\epsilon-1}$, the attacker reduces the loss to $1 - t^{\alpha+\epsilon-1}$. Otherwise, the attacker keeps the loss unchanged. In doing so, the attacker ensures that the loss of the target arm $\tilde{\mathcal{L}}_t(a^\dagger)$ is at least $t^{\alpha+\epsilon-1}$ smaller than $\tilde{\mathcal{L}}_t(a)$ for any non-target arm $a \neq a^\dagger$. As a result, $a^\dagger$ becomes the best-in-hindsight arm under $\tilde{\mathcal{L}}_{1:T}$. Note that the gap $t^{\alpha+\epsilon-1}$ diminishes as a function of $t$ since $\epsilon < 1 - \alpha$. The condition that $\epsilon$ must be strictly smaller than $1 - \alpha$ is important to achieving sublinear attack cost, which we will prove later.*

**Theorem 4.6.** *Assume the attacker applies (6) to perform attack. Then there exists a constant $M > 0$ such that the expected number of target arm selections satisfies*

$$\mathbf{E}\left[N_T(a^\dagger)\right] \geq T - \frac{1}{\alpha+\epsilon}T^{1-\alpha-\epsilon} - MT^{1-\epsilon}, \quad (7)$$

*and the expected cumulative attack cost satisfies*

$$\mathbf{E}\left[C_T\right] \leq \frac{1}{\alpha+\epsilon}T^{1-\alpha-\epsilon} + MT^{1-\epsilon} + \frac{1}{\alpha+\epsilon}T^{\alpha+\epsilon}. \quad (8)$$

**Remark 4.7.** *According to (7), the target arm will be selected more frequently as $\epsilon$ grows. This is because the attack (6) enforces that the loss of the target arm $\tilde{\mathcal{L}}_t(a^\dagger)$ is at least $t^{\alpha+\epsilon-1}$ smaller than the loss of non-target arms. As $\epsilon$ increases, the gap becomes larger, thus the bandit algorithm would further prefer $a^\dagger$. The cumulative attack cost, however, does not decrease monotonically as a function of $\epsilon$. This is because while larger $\epsilon$ results in more frequent target arm selections, the per-round attack cost may also increase. For example, if $\mathcal{L}_t(a^\dagger) = 1, \forall t$, then whenever $a^\dagger$ is selected, the attacker incurs attack cost $t^{\alpha+\epsilon-1}$, which grows as $\epsilon$ increases.*

**Corollary 4.8.** *Assume the attacker applies (6) to perform attack. Then when the attacker chooses $\epsilon = \frac{1-\alpha}{2}$, the expected cumulative attack cost achieves the minimum value asymptotically. Correspondingly, we have $\mathbf{E}\left[N_T(a^\dagger)\right] = T - O(T^{\frac{1+\alpha}{2}})$ and $\mathbf{E}\left[C_T\right] = O(T^{\frac{1+\alpha}{2}})$.*

We now show that our attack (6) recovers the results in Theorem 4.3 when it is applied in the easy attack scenario. We first provide another version of the theoretical bounds on $\mathbf{E}\left[N_T(a^\dagger)\right]$ and $\mathbf{E}\left[C_T\right]$ that depends on how close $\mathcal{L}_t(a^\dagger)$ is to the maximum value.

**Theorem 4.9.** *Let $\rho \in (0, 1]$ be any constant. Define $\mathcal{T}_\rho = \{t \mid \mathcal{L}_t(a^\dagger) > 1 - \rho\}$, i.e., the set of rounds where $\mathcal{L}_t(a^\dagger)$ is within distance $\rho$ to the maximum loss value. Let $|\mathcal{T}_\rho| = \tau$. Also assume that*

the attacker applies (6) to perform attack, then there exists a constant $M > 0$ such that the expected number of target arm selections satisfies

$$\mathbf{E}\left[N_T(a^\dagger)\right] \geq T - \rho^{\frac{1}{\alpha+\epsilon-1}} - \tau - MT^\alpha/\rho, \tag{9}$$

and the cumulative attack cost satisfies

$$\mathbf{E}\left[C_T\right] \leq \rho^{\frac{1}{\alpha+\epsilon-1}} + \tau + MT^\alpha/\rho. \tag{10}$$

**Remark 4.10.** *In the easy attack scenario, there exists some $\rho$ such that $\tau = 0$, thus compared to Theorem 4.3, the more generic attack (6) induces an additional constant term $\rho^{\frac{1}{\alpha+\epsilon-1}}$ in the bounds of $\mathbf{E}\left[N_T(a^\dagger)\right]$ and $\mathbf{E}\left[C_T\right]$, which is negligible for large enough $T$.*

## 5 ATTACK COST LOWER BOUND

We have proposed two attack strategies targeting the easy and general attack scenarios separately. In this section, we show that if an attack algorithm achieves $T - o(T)$ target arm selections and is also ***victim-agnostic***, then the cumulative attack cost is at least $\Omega(T^\alpha)$. Note that since we want to derive victim-agnostic lower bound, it is sufficient to pick a particular victim bandit algorithm that guarantees $O(T^\alpha)$ regret and then prove that any victim-agnostic attacker must induce at least some attack cost in order to achieve $T - o(T)$ target arm selections. Specifically, we consider the most popular Exp3 algorithm (see algorithm 1 in the appendix). We first provide the following key lemma, which characterizes a lower bound on the number of arm selections for Exp3.

**Lemma 5.1.** *Assume the bandit player applies the Exp3 algorithm with parameter $\eta$ (see (34) in the appendix) and initial arm selection probability $\pi_1$. Let the loss functions be $\mathcal{L}_{1:T}$. Then $\forall a \in \mathcal{A}$, the total number of rounds where $a$ is selected, $N_T(a)$, satisfies*

$$\mathbf{E}\left[N_T(a)\right] \geq T\pi_1(a) - \eta T \sum_{t=1}^T \mathbf{E}\left[\pi_t(a)\mathcal{L}_t(a)\right], \tag{11}$$

*where $\pi_t$ is the arm selection probability at round $t$. Furthermore, since $\pi_t(a) \leq 1$, we have*

$$\mathbf{E}\left[N_T(a)\right] \geq T\pi_1(a) - \eta T \sum_{t=1}^T \mathcal{L}_t(a). \tag{12}$$

**Remark 5.1.** *Lemma 5.1 provides two different lower bounds on the number of arm selections based on the loss functions for each arm $a$. (12) shows that the lower bound on $\mathbf{E}\left[N_T(a)\right]$ increases as the cumulative loss $\sum_{t=1}^T \mathcal{L}_t(a)$ of arm $a$ becomes smaller, which coincides with the intuition. In particular, if $\pi_1$ is initialized to the uniform distribution and $\eta$ is picked as $\beta T^{-\frac{1}{2}}$ for some constant $\beta$, the lower bound (12) becomes $\mathbf{E}\left[N_T(a)\right] \geq T/K - \beta\sqrt{T}\sum_{t=1}^T \mathcal{L}_t(a)$. One direct conclusion here is that if the loss function of an arm $a$ is always zero, i.e., $\mathcal{L}_t(a) = 0, \forall t$, then arm $a$ must be selected at least $T/K$ times in expectation.*

Now we provide our main result in Theorem 5.2, which shows that for a special implementation of Exp3 that achieves $O(T^\alpha)$ regret, any attacker must induce $\Omega(T^\alpha)$ cumulative attack cost.

**Theorem 5.2.** *Assume some victim-agnostic attack algorithm achieves $\mathbf{E}\left[N_T(a^\dagger)\right] = T - o(T)$ on all victim bandit algorithms that has regret rate $O(T^\alpha)$, where $\alpha \in [\frac{1}{2}, 1)$. Then there exists a bandit task such that the attacker must induce at least expected attack cost $\mathbf{E}\left[C_T\right] = \Omega(T^\alpha)$ on some victim algorithm. Specifically, one such victim is the Exp3 algorithm with parameter $\eta = \Theta(T^{-\alpha})$.*

The lower bound $\Omega(T^\alpha)$ matches the upper bound proved in both Theorem 4.3 and Theorem 4.9 up to a constant, thus our attacks are asymptotically optimal in the easy attack scenario. However, there is a gap compared to the upper bound $O(T^{\frac{1+\alpha}{2}})$ proved for the general attack scenario (corollary 4.8). The gap diminishes as $\alpha$ approaches 1, but how to completely close this gap remains an open problem.

## 6 EXPERIMENTS

We now perform empirical evaluations of our attacks. We consider two victim adversarial bandit algorithms: the Exp3 algorithm (see Algorithm 1 in the appendix), and a robust version of Exp3

called ExpRb (see Yang et al. (2020)). The ExpRb assumes that the attacker has a fixed attack budget $\Phi$. When $\Phi = O(\sqrt{T})$, the ExpRb recovers the regret of Exp3. However, our attack does not have a fixed budget beforehand. Nevertheless, we pretend that ExpRb assumes some budget $\Phi$ (may not be bounded by the cumulative attack cost of our attacker) and evaluate its performance for different $\Phi$'s. Note that as illustrated in example 1, robust bandit algorithms that can recover sublinear regret may still suffer from an attacker who aims at promoting a target arm in the adversarial bandit setting.

## 6.1 AN EASY ATTACK EXAMPLE

In out first example, we consider a bandit problem with $K = 2$ arms, $a_1$ and $a_2$. The loss function is $\forall t, \mathcal{L}_t(a_1) = 0.5$ and $\mathcal{L}_t(a_2) = 0$. Without attack $a_2$ is the best-in-hindsight arm and will be selected most of the times. The attacker, however, aims at forcing arm $a_1$ to be selected in almost very round. Therefore, the target arm is $a^\dagger = a_1$. Note that $\mathcal{L}_t(a^\dagger) = 0.5, \forall t$, thus this example falls into the easy attack scenario, and we apply (5) to perform attack.

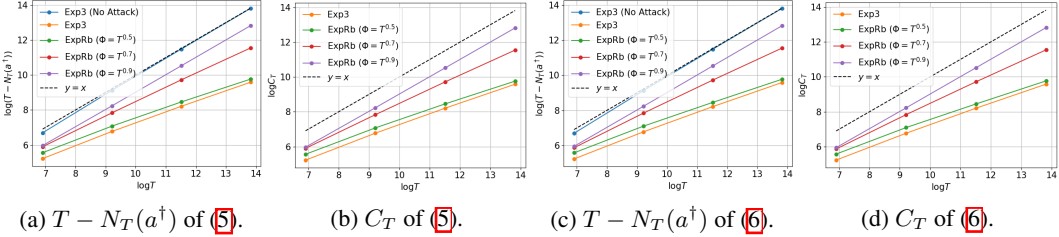

(a) $T - N_T(a^\dagger)$ of (5).    (b) $C_T$ of (5).    (c) $T - N_T(a^\dagger)$ of (6).    (d) $C_T$ of (6).

Figure 1: Using (5) and (6) to perform attack in an easy attack scenario.

In the first experiment, we let the total horizon be $T = 10^3, 10^4, 10^5$ and $10^6$. For each $T$, we run the Exp3 and ExpRb under attack for $T$ rounds, and compute the number of "non-target" arm selections $T - N_T(a^\dagger)$. We repeat the experiment by 10 trials and take the average. In Figure 1a, we show $\log(T - N_T(a^\dagger))$, i.e., the log value of the total number of averaged "non-target" arm selections, as a function of $\log T$. The error bars are tiny small, thus we ignore them in the plot. Smaller value of $\log(T - N_T(a^\dagger))$ means better attack performance. Note that when no attack happens (blue line), the Exp3 algorithm almost does not select the target arm $a^\dagger$. Specifically, for $T = 10^6$, the Exp3 selects $a^\dagger$ in $1.45 \times 10^4$ rounds, which is only $1.5\%$ of the total horizon. Under attack though, for $T = 10^3, 10^4, 10^5, 10^6$, the attacker misleads Exp3 to select $a^\dagger$ in $8.15 \times 10^2, 9.13 \times 10^3, 9.63 \times 10^4$, and $9.85 \times 10^5$ rounds, which are $81.5\%, 91.3\%, 96.3\%$ and $98.5\%$ of the total horizon. We also plotted the line $y = x$ for comparison. Note that the slope of $\log(T - N_T(a^\dagger))$ is smaller than that of $y = x$, which means $T - N_T(a^\dagger)$ grows sublinearly as $T$ increases. This matches our theoretical results in Theorem 4.3. For the other victim ExpRb, we consider different levels of attack budget $\Phi$. The attacker budget assumed by ExpRb must be sublinear, since otherwise the ExpRb cannot recover sublinear regret, and thus not practically useful. In particular, we consider $\Phi = T^{0.5}, T^{0.7}$ and $T^{0.9}$. Note that for $\Phi = T^{0.7}$ and $T^{0.9}$, the ExpRb cannot recover the $O(\sqrt{T})$ regret of Exp3. For $T = 10^6$, our attack forces ExpRb to select the target arm in $9.83 \times 10^6, 8.97 \times 10^6$, and $6.32 \times 10^6$ rounds for the three attacker budget above. This corresponds to $98.3\%, 89.7\%$, and $63.2\%$ of the total horizon respectively. Note that the ExpRb is indeed more robust than Exp3 against our attack. However, our attack still successfully misleads the ExpRb to select the target $a^\dagger$ very frequently. Also note that the attack performance degrades as the attacker budget $\Phi$ grows. This is because the ExpRb becomes more robust as it assumes a larger attack budget $\Phi$.

Figure 1b shows the attack cost averaged over 10 trials. For Exp3, the cumulative attack costs are $1.85 \times 10^2, 8.72 \times 10^2, 3.67 \times 10^3$, and $1.45 \times 10^4$ for the four different $T$'s. On average, the per-round attack cost is $0.19, 0.09.0.04$, and $0.01$ respectively. Note that the per-round attack cost diminishes as $T$ grows. Again, we plot the line $y = x$ for comparison. Note that slope of $\log C_T$ is smaller than that of $y = x$. This suggests that $C_T$ increases sublinearly as $T$ grows, which is consistent with our theoretical results in Theorem 4.3. For ExpRb, for $T = 10^6$, our attack incurs cumulative attack costs $1.73 \times 10^4, 1.03 \times 10^5$ and $3.68 \times 10^5$ when ExpRb assumes $\Phi = T^{0.5}, T^{0.7}$ and $T^{0.9}$ respectively. On average, the per-round attack cost is $0.02, 0.10$ and $0.37$. Note that our attack induces larger attack cost on ExpRb than Exp3, which means ExpRb is more resilient against

our attacks. Furthermore, the attack cost grows as ExpRb assumes a larger attack budget $\Phi$. This is again due to that a larger $\Phi$ implies that ExpRb is more prepared against attacks, thus is more robust.

Next we apply the general attack (6) to verify that (6) can recover the results of Theorem 4.3 in the easy attack scenario. We fix $\epsilon = 0.25$ in (6). In Figures 1c and 1d, we show the number of target arm selections and the cumulative attack cost. For the Exp3 victim, for the four different $T$'s, the attack (6) forces the target arm to be selected in $8.12 \times 10^2, 9.12 \times 10^3, 9.63 \times 10^4, 9.85 \times 10^5$ rounds, which is $81.2\%, 91.2\%, 96.3\%$ and $98.5\%$ of the total horizon respectively. Compared to (5), the attack performance is just slightly worse. The corresponding cumulative attack costs are $1.89 \times 10^2, 8.76 \times 10^2, 3.68 \times 10^3$, and $1.45 \times 10^4$. On average, the per-round attack cost is $0.19, 0.09, 0.04$ and $0.01$. Compared to (5), the attack cost is almost the same.

## 6.2 A GENERAL ATTACK EXAMPLE

In our second example, we consider a bandit problem with $K = 2$ arms and the loss function is $\forall t, \mathcal{L}_t(a_1) = 1$ and $\mathcal{L}_t(a_2) = 0$. The attacker desires target arm $a^\dagger = a_1$. This example is hard to attack because the target arm has the maximum loss across the entire $T$ horizon. We apply the general attack (6) to perform attack. We consider $T = 10^3, 10^4, 10^5$, and $10^6$. The results reported in this section are also averaged over 10 independent trials.

In the first experiment, we let the victim bandit algorithm be Exp3 and study how the parameter $\epsilon$ affects the performance of the attack. We let $\epsilon = 0.1, 0.25$ and $0.4$. In Figure 2a, we show the number of target arm selections for different $T$'s. Without attack, the Exp3 selects $a^\dagger$ in only $1.20 \times 10^2, 5.27 \times 10^2, 2.12 \times 10^3$, and $8.07 \times 10^3$ rounds, which are $12\%, 5.3\%, 2.1\%$ and $0.81\%$ of the total horizon. In Figure 2a, we show $\log(T - N_T(a^\dagger))$ as a function of $\log T$ for different $\epsilon$'s. Note that as $\epsilon$ grows, our attack (6) enforces more target arm selections, which is consistent with Theorem 4.6. In particular, for $\epsilon = 0.4$, our attack forces the target arm to be selected in $8.34 \times 10^2, 9.13 \times 10^3, 9.58 \times 10^4, 9.81 \times 10^5$ rounds, which are $83.4\%, 91.3\%, 95.8\%$ and $98.1\%$ of the total horizon. In Figure 2b, we show the cumulative attack cost. Note that according to corollary 4.8, the cumulative attack cost achieves the minimum value at $\epsilon = 0.25$. This is exactly what we see in Figure 2b. Specifically, for $\epsilon = 0.25$, the cumulative attack costs are $4.20 \times 10^2, 2.84 \times 10^3, 1.85 \times 10^4$, and $1.14 \times 10^5$. On average, the per-round attack cost is $0.42, 0.28, 0.19$ and $0.11$ respectively. Note that the per-round attack cost diminishes as $T$ grows. In both Figure 2a and 2b, we plot the line $y = x$ for comparison. Note that both $T - \mathbf{E}\left[N_T(a^\dagger)\right]$ and $C_T$ grow sublinearly as $T$ increases, which verifies our results in Theorem 4.6.

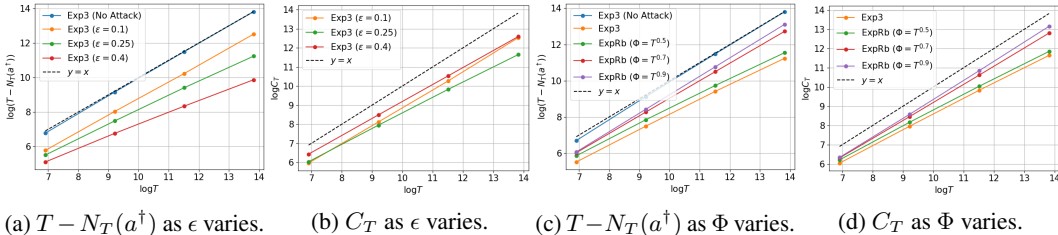

(a) $T - N_T(a^\dagger)$ as $\epsilon$ varies.  (b) $C_T$ as $\epsilon$ varies.  (c) $T - N_T(a^\dagger)$ as $\Phi$ varies.  (d) $C_T$ as $\Phi$ varies.

Figure 2: Using (6) to perform attack in general attack scenarios.

In our second experiment, we evaluate the performance of our attack (6) on the robust adversarial bandit algorithm ExpRb Yang et al. (2020). We fixed $\epsilon = 0.25$ in (6). We consider three levels of attacker budget $\Phi = T^{0.5}, T^{0.7}$ and $T^{0.9}$ in ExpRb, corresponding to increasing power of the attacker. In Figure 2c, we show the total number of target arm selections. For $T = 10^6$, our attack forces the Exp3 to select the target arm in $9.24 \times 10^5$ rounds, which is $92.4\%$ of the total rounds. For the ExpRb victim, for the three different attack budgets $\Phi$'s, our attack forces ExpRb to select the target arm in $8.97 \times 10^5, 6.65 \times 10^5$ and $5.07 \times 10^5$ rounds, corresponding to $89.7\%, 66.5\%$ and $50.7\%$ of the total horizon respectively. Note that when $\Phi = T^{0.5}$, i.e., the ExpRb can recover the regret of Exp3, but our attack still forces target arm selection in almost $90\%$ of rounds. This is smaller than the $92.4\%$ on the Exp3 victim, which demonstrates that ExpRb indeed is more robust than Exp3. Nevertheless, the ExpRb failed to defend against our attack. Even when ExpRb assumes a very large attacker budget like $\Phi = T^{0.9}$, our attack still forces the target arm selection in $50.7\%$ of rounds.

In Figure 2d, we show the cumulative attack costs. For $T = 10^6$, the cumulative attack cost on the Exp3 victim is $1.14 \times 10^5$. On average, the per-round attack cost is $0.11$. For the ExpRb victim, for $T = 10^6$, the cumulative attack costs are $1.40 \times 10^5, 3.62 \times 10^5$, and $5.15 \times 10^5$ for the three different attacker budgets $\Phi = T^{0.5}, T^{0.7}, T^{0.9}$. The per-round attack cost is $0.14, 0.36$ and $0.51$ respectively. Note that when $\Phi = T^{0.5}$, the ExpRb recovers the regret of Exp3. The per-round attack cost for ExpRb is $0.14$, which is slightly higher than Exp3. This again shows that ExpRb is indeed more robust than Exp3. Also note that the attack cost grows as ExpRb assumes a larger attacker budget. This is reasonable since larger attacker budget $\Phi$ implies stronger robustness of ExpRb.

## 7 RELATED WORKS

Existing research on attacks of multi-armed bandit mostly fall into the topic of data poisoning Ma et al. (2019b). Prior works are limited to poisoning attacks on stochastic bandit algorithms. One line of work studies reward poisoning on vanilla bandit algorithms like UCB and $\epsilon$-greedy Jun et al. (2018); Zuo (2020); Niss; Xu et al. (2021b); Ma et al. (2018); Liu & Shroff (2019); Wang et al. (2021); Ma (2021); Xu et al., contextual and linear bandits Garcelon et al. (2020), and also best arm identification algorithms Altschuler et al. (2019). Another line focuses on action poisoning attacks Liu & Lai (2020; 2021a) where the attacker perturbs the selected arm instead of the reward signal. Recent study generalizes the reward attacks to broader sequential decision making scenarios such as multi-agent games Ma et al. (2021) and reinforcement learning Ma et al. (2019a); Zhang et al. (2020); Sun et al. (2020); Rakhsha et al. (2021); Xu et al. (2021a); Liu & Lai (2021b), where the problem structure is more complex than bandits. In the multi-agent decision-making scenarios, a related security threat is an internal agent who adopts strategic behaviors to mislead competitors and achieves desired objectives such as Deng et al. (2019); Gleave et al. (2019).

There are also prior works that design robust algorithms in the context of stochastic bandits Feng et al. (2020); Guan et al. (2020); Rangi et al. (2021); Ito (2021), linear and contextual bandits Bogunovic et al. (2021); Ding et al. (2021); Zhao et al. (2021); Yang & Ren (2021); Yang (2021), dueling bandits Agarwal et al. (2021), graphical bandits Lu et al. (2021), best-arm identification Zhong et al. (2021), combinatorial bandit Dong et al. (2022), and multi-agent Vial et al. (2022) or federated bandit learning scenarios Mitra et al. (2021); Demirel et al. (2022). Most of the robust algorithms are designed to recover low regret even in presence of reward corruptions. However, as we illustrated in example 1, recovering low regret does not guarantee successful defense against an attacker who wants to promote a target arm in the adversarial bandit scenario. How to defend against such attacks remains an under-explored question.

Of particular interest to our paper is a recent work on designing adversarial bandit algorithms robust to reward corruptions Yang et al. (2020). The paper assumes that the attacker has a prefixed budget of attack cost $\Phi$, and then designs a robust adversarial bandit algorithm ExpRb, which achieves regret that scales linearly as the attacker budget $\Phi$ grows $R_T = O(\sqrt{K \log KT} + K\Phi \log T)$. As a result, the ExpRb can tolerate any attacker with budget $\Phi = O(\sqrt{T})$ while recovering the standard regret rate of Exp3. We point out that one limitation of ExpRb is that it requires prior knowledge of a fixed attack budget $\Phi$. However, our attack does not have a fixed budget beforehand. Instead, our attack budget depends on the behavior of the bandit player. Therefore, the ExpRb does not directly apply as a defense against our attack. Nevertheless, in our experiments, we pretend that ExpRb assumes some attack budget $\Phi$ and evaluate its performance under our attack.

## 8 CONCLUSION

We studied reward poisoning attacks on adversarial multi-armed bandit algorithms. We proposed attack strategies in both easy and general attack scenarios, and proved that our attack can successfully mislead any no-regret bandit algorithm into selecting a target arm in $T - o(T)$ rounds while incurring only $o(T)$ cumulative attack cost. We also provided a lower bound on the cumulative attack cost that any victim-agnostic attacker must induce in order to achieve $T - o(T)$ target arm selections, which matches the upper bound achieved by our attack. This shows that our attack is asymptotically optimal. Our study reveals critical security caveats in bandit-based applications, and it remains an open problem how to defend against our attacker whose attack goal is to promote a desired target arm instead of boosting the regret of the victim bandit algorithm.

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
