# OpenReview forum: "Adversarial Attacks on Adversarial Bandits"
_ICLR.cc/2023/Conference — ICLR 2023 notable top 25%_

### Official Review · Reviewer_2uWK · 2022-10-23

**Confidence:** 4
**Correctness:** 4
**Technical Novelty And Significance:** 4
**Empirical Novelty And Significance:** 3
**Recommendation:** 8

**Clarity, Quality, Novelty And Reproducibility:**

The paper is clear. The idea of manipulating an adversarial bandit is novel and very interesting. The theoretical results are well-presented and solid.

**Strength And Weaknesses:**

Strengths:

The paper is very well-written. The problem setting and assumptions used are all clearly described. The idea of manipulating an adversarial bandit is novel and very interesting. The theoretical results are well-presented. They are concise and easy to follow but tell a complete story at the same time.

Weaknesses:

The proposed methods require knowledge of the bandit algorithm's regrete rate in some cases.

**Summary Of The Paper:**

The paper studies the problem of how to manipulate an adversarial bandit algorithm. In the problem an attacker can perturb the loss function and wants to steer the bandit algorithm to select a target action of the attacker's choice. The authors presented methods to pertub the loss function to achieve this goal in two different settings, and showed that the methods incur only sublinear attack costs. Experiments were also conducted to evaluate the performance of the proposed methods.

**Summary Of The Review:**

Overall, a well-executed paper, consise and easy to follow. The key idea is novel and has the potential of inspiring follow-up work. The results are solid and coherent.

---

> ### Author Response · Authors · 2022-11-06
> **Response to Reviewer**
>
> We thank the reviewer for constructive and valuable comments.
>
> (1). The proposed methods require knowledge of the bandit algorithm's regret rate in some cases.
>
> Our attack strategy in the easy attack scenario (equation (5)) does not require knowledge of the regret rate. However, the general attack strategy (equation (6)) does require knowledge of the regret rate. At least, the attacker needs to know an upper bound on the regret rate. In appendix A.2, we provided some analysis on how the attack performance changes if the attacker assumed some regret rate different from the true regret rate.

---

### Official Review · Reviewer_dVy5 · 2022-10-24

**Confidence:** 4
**Correctness:** 3
**Technical Novelty And Significance:** 2
**Empirical Novelty And Significance:** 2
**Recommendation:** 6

**Clarity, Quality, Novelty And Reproducibility:**

The clarity of the presentation is good. The quality of writing and organization is great, but in terms of technical contribution, the paper is not tackling a challenging research question. The paper is novel in the sense that it is the first that designs attacks for adversarial bandits.

**Strength And Weaknesses:**

**Strengths:**
- The paper is well-written and easy to follow.
- The topic of bandits with corruption is a timely topic.
- The motivation for adversarial bandits with corruption is less clear, but it is well justified in this paper.
- The proposed attacks are easy to understand and intuitive.

**Weaknesses:**
- This paper has a foundational issue with the way that the cost associated with the attacker is calculated. Consider the easy attack setting. The attacker will set the loss of all arms but the target arm to 1. This means that the attacker has to pay at most $(K-1)\times T$ for corrupting the loss of all other arms. However, the authors simply ignore this huge cost and calculate the cost only for the selected arm. This is the case for the general attack case also. If this observation is correct (I urge the authors to clarify this in rebuttal), the contribution of this paper is not interesting at all since the attacker is paying a huge cost of changing the entire game in their favor, but the authors just claim this is still good (sublinear cost) since only they count the cost when a corrupted arm is selected.

- Beyond the above major issue, I think the contribution of this paper is interesting but not surprising in terms of the attack design. The attack idea is very straightforward, i.e., set the loss of all arms except the target arm to 1. Nothing intellectually interesting here.



**Summary Of The Paper:**

This paper develops some attack strategies for adversarial bandits. They design two attack policies that need a sublinear cost of corrupting the original loss of arms and resulting in a linear regret for the bandit algorithm. On the lower bound side, the authors show their attack policy is optimal in the sense that there is no better attack policy with less cost that can degrade the regret of the bandit algorithms. Last, experimental results show that the proposed attack policy can degrade the regret of the existing bandit algorithms.

**Summary Of The Review:**

Overall, I found this paper borderline. My major issue is that the attack policies are very straightforward to design, which might be ok since this is the first paper introducing attacks for adversarial bandits. However, I would like to see how convincing is the response from the authors regarding the cost associated with the attackers.

---

> ### Author Response · Authors · 2022-11-06
> **Response to Reviewer**
>
> We thank the reviewer for constructive and valuable comments.
>
> (1) Attack cost definition
>
> First of all, we want to point out that our threat model and attack cost definition are in fact exactly the same as the seminal works on this topic (see [1] and [2] below).
>
> Note that in our threat model, the attacker does "not" reset the entire loss function, but only perturbs the instantiated loss value observed for the selected arm a_t. This is described in section 3 attack problem formulation. We noticed that the confusion may arise from our definition of attack cost (i.e., equation (3)), where we wrote the perturbed loss value \tilde \ell_t(a_t) as a function of the selected arm a_t. However, the definition should really be |\tilde \ell_t-\ell_t| instead, where only the instantiated loss value is considered. We apologize for this confusion and have updated the definition.
>
> The confusion may also arise from the way that we design the attack algorithms. In our paper, the attacker "imagines" a different loss function "in mind", which we call a template loss function. Note that our attacker does "not" directly set the environmental loss function to the template loss function, but only perturbs the instantiated loss value of the selected arm according to the template. In other words, this template loss function is never realized but only serves as a guidance for the attacker to decide how to perturb the loss value of the selected arm. That means our attack has the same threat model as [1] and [2]. Importantly, we show that due to the partial observability property in bandits, perturbing the instantiated loss value for the selected arm creates an illusion for the bandit player that it is learning under the template loss functions (see our key Observation 1 on page 4). As a result, the standard adversarial bandit regret guarantee would apply with respect to the template loss functions.
>
> In summary, in both our threat model and the attack algorithms, the attacker does "not" set a new loss function completely, but only perturbs the loss value of the selected arm, thus our attack cost also only considers the selected arm.
>
> (2). The attack design is not interesting
>
> Although the attack design is simple for the easy attack scenario, it is really not simple for the general attack scenario, where the target arm can have the maximum loss value over all time steps. We propose general attack algorithms with theoretical guarantee, which is technically nontrivial. Furthermore, we provided theoretical lower bound for the attack cost, which is an interesting impossibility result that helps guide the future work.
>
> [1] Adversarial Attacks on Stochastic Bandits (NeurIPS 2019)
>
> [2] Data Poisoning Attacks on Stochastic bandits (ICML 2020)

---

### Official Review · Reviewer_Vf3c · 2022-10-26

**Confidence:** 3
**Correctness:** 4
**Technical Novelty And Significance:** 3
**Empirical Novelty And Significance:** 3
**Recommendation:** 6

**Clarity, Quality, Novelty And Reproducibility:**

Writing and clarity is clear, I have no issues.
Some math proof sketched in the main paper would be better, as I am not sure if the math is hard or easy.
The idea appear novel to me.


**Strength And Weaknesses:**

This is primarily a theory paper. The experiments are in simulation. Overall results look fine; for a theory paper the theory is not too much of an advance but ok.

The attacks are straightforward enough, essentially making all other arms unattractive. But I am left wondering what would happen (theoretically) if a robust algorithm was being used, meaning the defender is aware of an attacker of the type in this paper - would the result of the attack be different? My view is possibly not, because the attacker has no much power here as to arbitrarily modify the reward (with a liberal cost allowance that can grow with T), which in my view also makes the attacker's task simple.
Is the concept of regret w.r.t. to the original rewards meaningful here? (meaning the defender observes changed rewards but collects the true reward in practice) If meaningful, what happens to this original regret value?


It would have been easier to see the graphs if the quantities were plotted averaged over T. It would have been nice if the dependence on the number of arms was also included in the formulas of the results. (which can be extracted from the regret guarantee).

Typos:
Example 1 - the word "contract" should be contradict

**Summary Of The Paper:**

The authors study a threat to adversarial multi-armed bandit, in which an attacker perturbs the loss or reward signal to force the defender to select a suboptimal target action in every but (T −o(T)) number of rounds, while incurring only o(T) cumulative attack cost. This result try to motivate the problem through online recommendation. The proposed attack algorithms require knowledge of only the regret rate, and is agnostic to the bandit algorithm used by the victim player. A theoretical lower bound on the cumulative attack cost tis also presented.

**Summary Of The Review:**

I think the work is novel, but not a big advance.

---

> ### Author Response · Authors · 2022-11-06
> **Response to Reviewer**
>
> We thank the reviewer for your constructive and valuable comments.
>
> (1) What would happen (theoretically) if a robust algorithm was being used?
>
> We agree with the reviewer that even if a robust algorithm was used, our attack is still effective and efficient. This is because most existing robust bandit algorithms aim at recovering sublinear regret in the presence of reward/loss corruptions. However, our attack does not directly aim at boosting the regret of the bandit player, but instead forcing the bandit player to always select a target arm. Note that these two goals do not necessarily align with each other (see our Example 1), thus traditional robust bandit algorithms may not provide effective defense against our attack. In fact, our attack works for any bandit algorithm that satisfies Assumption 2.2, including those traditional robust bandit algorithms. Therefore, theoretically all of our results still hold even if a robust algorithm was used. Empirically, we evaluated the performance of ExpRb, which is a robust version of Exp3, and the result shows that our attack is also successful on ExpRb.
>
> To defend against our attack, we believe the bandit algorithm needs to possess certain fairness properties. Intuitively, even if the target arm is promoted to be the best arm by the attacker, the bandit algorithm will not always select the target arm due to fairness concern.
>
> (2). Is the concept of regret w.r.t. to the original rewards meaningful here?
>
> In general, the regret with respect to the original rewards under our attack is no longer meaningful and can be linear in the total rounds T. However, there are special cases where the regret with respect to the original rewards remains sublinear. For example, in our Example 1, even if the target arm is selected in every round, the total regret with respect to the original rewards is only sqrt(T).
>
> (3). It would have been easier to see the graphs if the quantities were plotted averaged over T.
>
> We thank the reviewer for the nice suggestion. In the current plot, we let the x axis be log(T) and the y axis be the log value of the quantities in order to show that the quantities grow sublinearly as T increases. It's easier to show that relationship in log scale, especially when we want to see the slope. Because of that, averaging the quantities over T just rotates the current plot clockwise by some degrees, thus we did not average over T for simplicity. We plot the function y=x in dotted lines for comparison. We see that the slope of all the lines under attack are smaller than y=x, which demonstrates the sublinearity.
>
> (4). It would have been nice if the dependence on the number of arms was also included in the formulas of the results.
>
> This is a great comment. We provided more explanations on this question in the end of remark 4.4. Specifically, our paper made a very loose assumption about the victim bandit algorithm (assumption 2.2), which only requires the regret to be sublinear. The constant term in the regret bound may take different forms for different algorithms. There might be super efficient bandit algorithm such that the constant term M in the regret bound does not depend on the number of arms K at all. In this case, our theoretical bounds also do not depend on K. Therefore, technically we cannot spell out the concrete form of M based on K. Comparatively, the sublinear regret rate alpha is more important for attack considerations. Nevertheless, for the standard EXP3 algorithm, M scales as sqrt(K*log(K)), and our results can be instantiated accordingly.

---

### Decision · Program_Chairs · 2023-01-20

**Decision:**

Accept: notable-top-25%

**Justification For Why Not Higher Score:**

Reviewers overall liked the paper and the contribution, though they also find some of the results not that surprising.

**Justification For Why Not Lower Score:**

Reviewers overall liked the paper and the contribution, though they also find some of the results not that surprising.

**Metareview: Summary, Strengths And Weaknesses:**

The paper studies adversarial bandits, a setting in which an attacker can alter the reward received by a MAB to alter their behavior, a topic that has recently received attention. The main contribution of the paper is that a sublinear (o(T)) attack "budget" suffices to yield suboptimal target actions in T-o(T) rounds. Interestingly, this can be accomplished even though the attacker is assumed to have quite limited knowledge; in particular, the attacker only knows the regret rate but not the specific MAB algorithm used. A lower bound is that matches the strength of the attack is also provided.

The authors are encouraged to include the results on robust MABs to the paper: both the experiments, and the a discussion as to why the attack is still valid even in the presence of a robust MAB algorithm, would improve the overall contribution. The same is true about the dependence on arm cardinality, and the meaning of original rewards.

**Note From Pc:**

if the above contains the word "oral" or "spotlight" please see: "oral" presentation means -> notable-top-5% and "spotlight" means -> notable-top-25%. As stated in our emails, we are disassociating presentation type from AC recommendations

**Summary Of Ac-Reviewer Meeting:**

N/A